# The Associations between Physical Fitness, Complex vs Simple Movement, and Academic Achievement in a Cohort of Fourth Graders

**DOI:** 10.3390/ijerph18052293

**Published:** 2021-02-26

**Authors:** Jong-Sik Ryu, Hae Ryong Chung, Benjamin M. Meador, Yongsuk Seo, Kyung-O Kim

**Affiliations:** 1Department of Physical Education, Kyungpook National University, Daegu 41566, Korea; jsryu1984@gmail.com; 2Health and Fitness Management, College of Health, Clayton State University, Morrow, GA 30260, USA; hchung@clayton.edu; 3Exercise Science, College of Nursing and Health Sciences, Georgia Southwestern State University, Americus, GA 31709, USA; Benjamin.meador@gsw.edu; 4 Environmental Physiology Laboratory, Kent State University, Kent, OH 44242, USA; yseo@kent.edu; 5Department of Gerokinesiology, Kyungil University, Kyungsan 38428, Korea

**Keywords:** academic achievement, elementary school student physical fitness, complex movement, simple movement

## Abstract

This study analyzed the correlation between elementary school students’ body composition, physical activity, physical fitness, movement ability, and academic achievement. Movements ranged from simple actions to complex movements requiring executive functioning. In total, 110 fourth graders (60 boys, 50 girls) participated in this experiment. Body composition (BMI, % of body fat), physical activity (pedometer), physical fitness (muscular strength, endurance, power, flexibility, and VO_2_max), and complex movement abilities (Illinois Agility test, soda pop hand test, and soda pop foot test) were measured. Regression modeling of body composition and fitness/activity variables was able to account for 30.5% of the variation of total academic scores in females, but only 4.3% in males. No individual tests were reliably correlated with multiple academic outcomes in males. However, hand and foot soda pop times, as well as Illinois Agility scores, were repeatedly correlated with academic outcomes in females, each correlating with 4 of the 6 academic scores. Body composition and physical activity level did not correlate with academic achievement, and simple physical fitness showed a low correlation with academic achievement in both boys and girls. On the other hand, complex, cognitively demanding movements such as the Illinois Agility, soda pop hand, and soda pop foot tests had consistent correlations with academic achievement in girls, but not in boys.

## 1. Introduction

It is well established that insufficient physical activity (PA) is associated with adverse health outcomes such as diabetes, obesity, hypertension, coronary heart disease, osteoporosis, colon cancer, breast cancer, and even some psychiatric disorders [1]. Furthermore, regular physical activity improves cardiovascular and muscular fitness, bone health, psychological outcomes, and cognitive and brain health [2]. Thus, regular physical activity is crucial to improving overall health and reducing risks for many chronic diseases across the life span, as well as mortality. However, social distancing guidelines during the COVID-19 pandemic have changed lifestyles and limited leisure time activities, which decreases overall physical activity. This new social issue extends to children as well, who are seeing decreased activity in general, as well as loss of school-based physical education and extracurricular activities. While the consequences will take some time to become fully apparent, these changes are expected to negatively affect children’s physical and psychological health [3,4].

A previous study reported that physical activity habituation in childhood is related to PA habits in subsequent life stages and correlates with healthier lifestyles in adulthood [5]. PA in school-age youth promotes growth and motor development, which is related to vision, perception, and nervous system development [6,7,8]. 

Additionally, positive relationships between physical activity and cognitive function in children have been reported in multiple examinations [9], and subject-specific relationships between PA and academic achievement in children have also been indicated [10]. Specifically, in the 4–18-year-old range, meta-analysis has reported positive relationships between PA and cognitive function, perceptual skills, the intelligence quotient, verbal tests, mathematic tests, memory, developmental level, and academic readiness [11]. PA can increase cerebral blood flow and arousal level, alter brain neurotransmitters and central nervous system structure [12], and stimulate cognitive development [11,12]. Several studies have reported a significant relationship between physical fitness and academic achievement in elementary school students [10,13]. Specifically, Castelli, Hillman, Buck, and Erwin [13] reported that academic achievement is positively correlated with aerobic capacity and negatively correlated with body mass index (BMI). Regular physical activity has been shown to have a moderate positive effect on academic achievement [10], though this has not always been reported consistently [14]. One likely confounder to the consistency of this literature is that approaches to physical activity, physical fitness, and academic testing are not standardized. The fact that most activity assessment batteries are primarily designed for adults and may not be valid in younger populations makes this especially problematic in school-aged children [15]. 

Associations between physical activity and cognitive performance might be attributed to the benefits of daily PA on developing neural networks [16,17]. Further, participation in PA may change capacities of neural networks, such as the default mode, executive function, and motor function [17]. Previous studies have also demonstrated that physical fitness is related to learning and memory [18]. Blair and Razza [19] observed that the executive function was positively related to mathematics skills and emerging literacy. Cognitive functions such as attention, information processing, executive function, and memory are crucial for academic achievement. 

Although many studies suggest that PA is related to academic achievement, they are limited to aerobic exercise, the amount of physical activity, and physical fitness and body composition [13,20,21]. A previous study reported that endurance running, push-ups, and curl-ups were positively correlated with academic achievement (total of mathematics and reading achievement) [13]. Another previous work reported that higher levels of moderate to vigorous physical activity and less sedentary time were related to better reading skills in grades 2 and 3 [20].

It is important to additionally explore other potential factors in the relationship between academic achievement and physical capacities. Therefore, the purpose of the current study was to examine the associations between academic achievement and complex movement tasks such as the Illinois Agility Test, and whether such tests can illuminate additional interactions beyond fitness/BMI/PA, etc. It was hypothesized that academic achievement (Korean literacy, mathematics, social studies, science, and English) would be positively correlated with physical activity (VO_2_max, sit-ups, sit-and-reach, standing long jump, and hand grip), and inversely correlated with BMI and performance times for the 50-m run, Illinois Agility test, and hand and foot soda pop tests. However, the correlations with complex tasks requiring higher executive functioning (i.e., planning and remembering/following instruction)—such as the Illinois agility and soda pop tests—were expected to show the strongest associations with academics.

## 2. Materials and Methods

The fourth-grade class of a South Korean elementary school—136 students in total—was recruited for participation in this study. After students and parents received an explanation of the study and its purpose, 3 declined to participate, and the remaining 133 were enrolled with parental consent. Subsequently, 5 students transferred to other schools, and 18 were ultimately excluded from the dataset due to incomplete participation. In total, 110 (60 boys, 50 girls) subject datasets were completed. The research was approved by the Kyungpook National University, South Korea (#2019-0135). The subject characteristics are shown in Table 1. The anthropometric/fitness/PA measurements of this study were conducted in the second semester of the fourth grade, and academic achievement scores reflect the final scores at the end of that semester and therefore, also the end of the fourth grade.

Collected anthropometrics included height, weight, and BMI (BSM 330, Biospace, Seoul, Korea), and percentage of body fat (skin folds) [22]. Tests of simple physical fitness were performed in accordance with and as part of the Korean Ministry of Education’s Physical Activity Promotion System (PAPS), which included muscular strength by a hand grip dynamometer, muscle endurance by a sit-up test [23], standing long jump, and sit-and-reach. Additionally, VO_2_max was assessed by the Bruce protocol (Quark b2 Metabolic Cart, Rome, Italy). Complex movement tasks included the Illinois agility test, soda pop hand test [24] and soda pop foot test, as described below. Collection of movement-task data was done over an approximately one-week period to avoid interference with the regular curriculum.

Physical activity was collected by a pedometer. PE-105 (Sinwoo Electronics, Siheung-si, Korea) pedometers were distributed to the students, and instructions were given on proper wearing and use. To minimize interaction with other measures, pedometer collection was done after the administration of PAPS tests. As supported by the work of Tudor-Locke and Myers [25], students wore pedometers for seven days during all activities of daily life, including weekends, except while sleeping or bathing. There were no rainy days during the measurement period of physical activity using pedometers. Pedometers were worn at the intersection of the midline of the waist and thigh, and counts were collected every morning to minimize the number of people who failed to collect data.

The soda pop hand test was performed according to the original AAHPERD (American Alliance for Health, Physical Education, Recreation and Dance) protocol [26]. This test was chosen for its combination of coordinative and attentional/cognitive demands, while having a low requirement for general physical demands (i.e., strength). Briefly, if testing the right hand, the participant started with their hand on can A (Figure 1), thumb-up. On the start signal, can A was flipped over to rest upside-down in circle 2, then the same was done with can B into circle 4 and can C into circle 6. The participant then returned to can A, flipping it back upright to its original position in circle 1, and the same was done with cans B and C. This cycle was repeated two additional times. The participants were allowed two practice trials, then two actual trials. The best score was recorded to the closest one-tenth of a second.

The soda pop foot test involved moving the soda cans by foot from a standing position, one at a time, as fast as possible. To perform this test, the cans and target circles were set on the floor instead of a table. The overall test follows the same move-A-B-C, return A-B-C pattern of the standard soda pop test, with the exception that the cans were not flipped over, but simply slid into position instead. The cycle was repeated one additional time, so that each can was moved and returned twice. The test was performed with the dominant foot, and times were recorded to the nearest tenth second.

Data were analyzed in SPSS (IBM, v.23, Armonk, NY, USA). Significant differences between groups (male vs female) were examined by *t*-test, and correlations between body composition, activity, and fitness tests were examined by Pearson correlations. 

In addition to correlational comparisons, multi-predictor regression models for total grade scores were generated through the automatic linear modeling function of SPSS Statistics. Separate models were created for males and females, built through the “best subsets” predictor selection method. Automatic data preparation for screening of outliers (>3 SD from the mean) was enabled, and BMI, wait hip ratio (WHR), and all activity, fitness, and coordination test variables were included as possible predictors.

## 3. Results

### 3.1. Correlations

PA and fitness measures are shown in Table 2. On average, PA by pedometer, sit-up repetitions, distance long jump, and Illinois Agility times showed a significantly higher performance in males than females, while the females attained significantly better sit-and-reach scores.

In males, physical activity was correlated with Korean Literature (r = 0.295, *p* = 0.022), and the soda pop hand test times were correlated with math scores (r = −0.260, *p* = 0.045). In females, sit-and-reach scores correlated with English (r = 0.295, *p* = 0.038), while the soda pop hand test times correlated with math (r = −0.297, *p* = 0.036), science (r = −0.287, *p* = 0.043), English (r = −0.411, *p* = 0.003), and total grades (r = −0.413, *p* = 0.003). Soda pop foot times correlated with social studies (r = −0.404, *p* = 0.004), science (r = −0.320, *p* = 0.023), English (r = −0.341, *p* = 0.015), and total grades (r = −0.409, *p* = 0.03). Illinois agility times correlated with Korean literature (r = −0.287, *p* = 0.043), social studies (r = −0.352, *p* = 0.012), science (r = −0.373, *p* = 0.008), and total grades (r = −0.313, *p* = 0.027). Significant correlations are shown in Figure 2.

### 3.2. Regression Modeling

In females, the regression model for total grade scores was able to explain 30.5% of the variance. In total, 3 of the 50 subjects were excluded from the model as outliers. In order of importance, the included predictors were soda pop foot times, physical activity, Illinois agility, sit-and-reach scores, and VO_2_max (Figure 3).

In males, the regression model for total grade scores was able to explain 4.3% of the variance. In total, 1 of the 60 subjects was excluded from the model as an outlier. Soda pop hand scores were the only contributor to the model; no other activity tests significantly improved the model fit (Figure 4).

## 4. Discussion

The current study examined the relationship between academic achievement (Korean literacy, mathematics, social study, science, English, and total grades) and fitness and activity-related predictor variables (amount of physical activity, BMI, VO_2_max, sit-ups, sit-and-reach, standing long jump, 50-m run, hand grip, Illinois agility test, and soda pop tests) in fourth-grade students. 

While repeated correlations were found between girls’ soda pop hand, soda pop foot, and Illinois Agility times and various academic scores, these relationships were markedly different in boys. Only passing correlations were found in boys, between PA and Korean Literature, and the soda pop-hand test and math.

In the current findings, academic achievement of overweight or obese fourth-grade students was similar to their peers with average weights, which is in agreement with previous findings [27]. However, one previous study reported that an overweight status in girls was related to lower academic achievement, though the same was not found in boys [28]. 

In the current dataset, no correlations were found between simple fitness tests (physical activity, BMI, sit-ups, sit-and-reach, standing long jump, 50-m run, and hand grip) and academic achievement, except for the correlations between sit-and-reach and English scores in girls. This is largely contrary to our hypothesis and also contrasts with some previous results. Cross-sectional studies using FITNESSGRAM^®^ (Cooper Institute, Dallas, TX, USA) for fitness assessment have reported a positive relationship between physical fitness and academic achievement [29,30,31,32,33]. Longitudinal studies also reported associations between fitness and academic achievement [34,35,36]. The difference between the present study and previous studies, both cross-sectional and longitudinal, may be due to the assessment tools used for physical fitness and academic achievement. For example, three longitudinal studies applied WESTEST [37] and California standardized tests in mathematics, English [38], and literacy and numeracy [39]. 

Previous studies have reported potential benefits of physical activity on cognitive function, learning, and brain structure, which could influence academic success [37]. However, similar to the simple fitness tests, we noted only one correlation between physical activity and academics—that between PA by step-count and Korean literature in boys. This is perhaps unsurprising, as a review by Rasberry, Lee, Robin, Laris, Russell, Coyle, and Nihiser [14] analyzed 43 articles and reported mixed conclusions as to the relationship between PA and academic performance, finding an approximately even split between studies that found a relationship and those that did not. In that review, they did not find detrimental effects of physical inactivity on academic achievement [14]. 

The strongest and most consistent findings in the current dataset are those between complex motor/fitness tests and academics; hand and foot soda pop tests, as well as the Illinois Agility test, were significantly correlated with academic achievement in girls, though with only a passing correlation in boys. The cross-sectional nature of the current dataset does not allow the examination of this relationship’s directionality, and these tests are likely to be more sensitive than, for example, VO_2_max testing, to academic capacities affecting the physical test, rather than vice versa. In any case, this correlation between a coordinative test requiring executive function and academic performance appears to be novel in the literature. Significant further research is needed to elucidate the mechanisms of this relationship, should further literature confirm its existence. Examination of previous research concerning coordination reveals that motor coordination has been strongly correlated with the ability to maintain attention in school-aged children [40]. However, the current data set contains no measure for attentional ability, and so, it is not at all clear whether the current correlation between coordination and academic achievement is mediated through this or some entirely separate mechanism.

Another previous study reported that agility is related to spatial working memory and attention in preschool children [39]. Working memory ability has been shown to link to literacy and numeracy in 4–6-year-old children [40] and 11–12-year-old children [41]. These data on working memory bracket the ages of the current subjects (~9 years old). If the current academic data can stand as a proxy to those literary and numeric abilities, the current Illinois Agility test data in girls supports this relationship to agility found in preschoolers and indicates that it also extends to the older fourth-grade students examined here. However, it should be noted that—similar to coordination and attention—the current dataset does not have the measures to directly correlate agility and working memory.

Overall, a number of possible links seem to exist between the complex motor/fitness tasks performed here and academic achievement. Agility has been correlated with working memory [39], coordination has been correlated with attentional ability [42], and neural network capacities, learning, and memory have been shown to be modified by PA and fitness [17,18]. Further, second-grade students have shown enhanced attention and greater mathematical fluency after 5-min moderate-to-vigorous physical activity [42]. One review article suggests that physical activity can influence important components of academic performance, including cognitive skills, attitude, and classroom behavior [43]. In general, the current work sought to examine the question of what might be predictive of academic achievement; true mechanistic links between the test and academics are beyond the current dataset’s ability to address, and a great deal of further work is needed in this regard.

Of note from the current results is the clear difference seen between the male and the female cohorts; consistent correlations were found between the complex motor/fitness tasks and achievement in females, but not in males. Hints as to the possible bases of this difference are sparse in the literature. While a gender disparity has been reported in regard to the relationship between weight status and academic achievement [28], we have been unable to find previous reports of disparities in relationship to the types of tasks correlating with academics in the current dataset. It is worth noting, however, that there were significant differences in a number of measures across genders; namely, the boys showed higher PA and higher performance in sit-ups, long jump, and Illinois Agility, while the girls tested better in the sit-and-reach.

These gender disparities are not entirely unexpected, as differences in activity at similar ages have been previously reported [43,44,45], and PA in boys appears to increase even more in boys with development than it does in girls [46,47]. The different periods of secondary sexual characteristic development for boys (~12–13 years) and girls (~10–11 years) [48] suggest the likelihood that more girls than boys are reaching this developmental age in the current dataset, which may be beginning to drive such differences. Nevertheless, this remains speculative, as currently, there seem to be no datasets contrasting the effects of PA or cognitive functions on academic achievement in girls vs boys.

A reasonable initial hypothesis built from the current dataset might be that there is a sort of threshold effect; that PA/physical fitness beyond a certain level does not confer additional academic benefit. With a greater number of the male cohort being beyond such a threshold, the trend lines could break down, and significant relationships might not be found.

However, several points render this hypothesis unsatisfactory. One is that boys showed significantly higher PA levels than girls and did correlate PA with an academic outcome, while girls did not. Similarly, girls showed significantly higher sit-and-reach performance than boys, yet girls correlated sit-and-reach with an academic measure, while boys did not. This argues against any threshold/ceiling effect for these two measures. 

The only test that might make a case for a threshold effect seems to be the Illinois Agility test, in which boys outperformed girls, and only girls showed a relationship between the test performance and academics. The two further measures in which boys outperformed—long jump and sit-ups—did not correlate to academics in either gender and also do not appear to inform this hypothesis in either direction. Therefore, with only one fitness test supporting a possible threshold effect, it is difficult to lean on it for explanation.

Beyond the two-factor correlations, the regression analyses bring additional complexity to this question. The model for females was able to account for 30.5% of the variance, in contrast to only 4.3% for the males. The female model included soda pop foot times, VO_2_max, physical activity, Illinois agility, and sit-and-reach scores. The first two of these, VO_2_max and soda pop test times, were not significantly different between genders, yet they accounted for 42% of the overall predictive power of the model. Of the remaining three contributors, gender performance was mixed; boys outperformed in PA and Illinois Agility, but girls outperformed in the sit-and-reach. Overall, the correlations and regression models seem tenuous at best in rendering an explanation by the floor/ceiling affect.

A final point to examine in regard to the gender differences seen herein is that—in the currently examined age range—previous reports have shown significantly lower (as much as 44%) hand–eye coordination ability in girls than in boys [43,49]. However, soda pop test times did not show gender differences in the current dataset, indicating that such differences were not detectable in the current cohort. Examining the scatterplots for the significant soda pop tests/academic correlations (Figure 2, plots D through K) is interesting; the fastest test times correlate fairly consistently with higher academic scorings, with far less variation than tends to be seen at the slower test times. This raises the possibility that these correlations are at least partly driven by a highly coordinated, highly achieving subject subset, which—according to the previously referenced work—may not be typical. More work is needed to establish whether the current gender discrepancies are a persistent finding.

The current study is limited by several factors. First, this study analyzed only a single cohort of fourth-grade children from a single school, which may not be representative of other ages or other schools. Therefore, future studies are needed to determine if the present results extend to broader age groups. Additionally, physical activity was assessed only by pedometer; it might be beneficial to assess various types of PA, since different types of PA have shown different effects [10,50]. Finally, the cross-sectional nature of the current dataset does not allow cause-and-effect examination. Of particular interest in this regard would be the directionality of the academic relationship to complex tasks, to see if physical tasks with heavier cognitive loads also lead to greater cognitive gains than physical activity with a low cognitive demand.

## 5. Conclusions

The current study indicated that performance of complex movement tests was strongly related to academic achievement in girls, while simple fitness tests failed to correlate. Body composition and physical activity levels were not related to academic achievement in fourth-grade students. Future studies are needed to elucidate this relationship between complex movement skills and academic achievement, as well as the question of whether gender is truly a robust modifier of this relationship, as was indicated by the current results.

## Figures and Tables

**Figure 1 ijerph-18-02293-f001:**
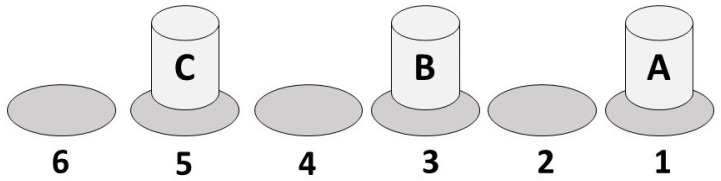
Initial arrangement of the cans and target circles for the soda pop tests, both hand and foot.

**Figure 2 ijerph-18-02293-f002:**
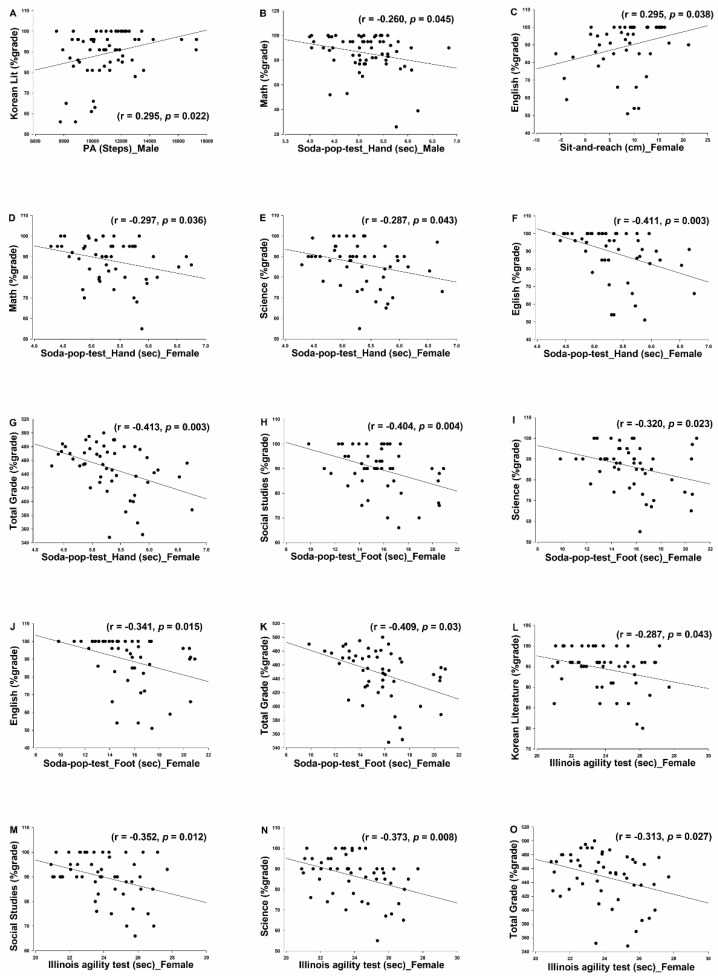
Scatterplots for the correlations that showed significant correlations between anthropometrics/PA/fitness tests and academic achievement scores.

**Figure 3 ijerph-18-02293-f003:**
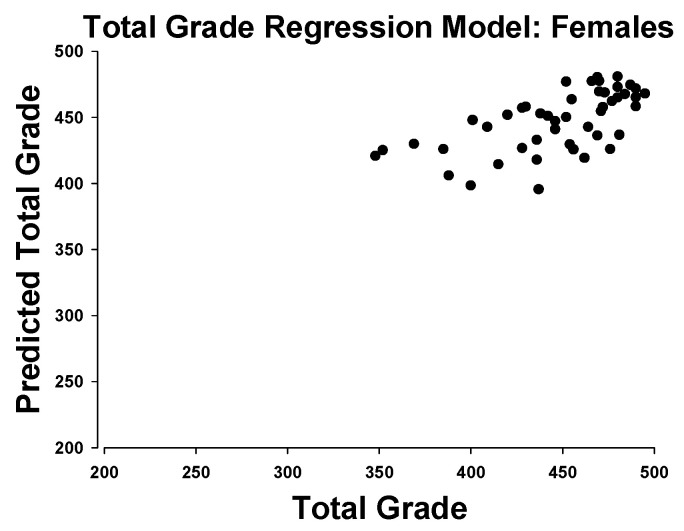
Actual total grade scores in females vs the predicted value from a regression model including soda pop foot times, physical activity, Illinois agility, sit-and-reach scores, and VO_2_max.

**Figure 4 ijerph-18-02293-f004:**
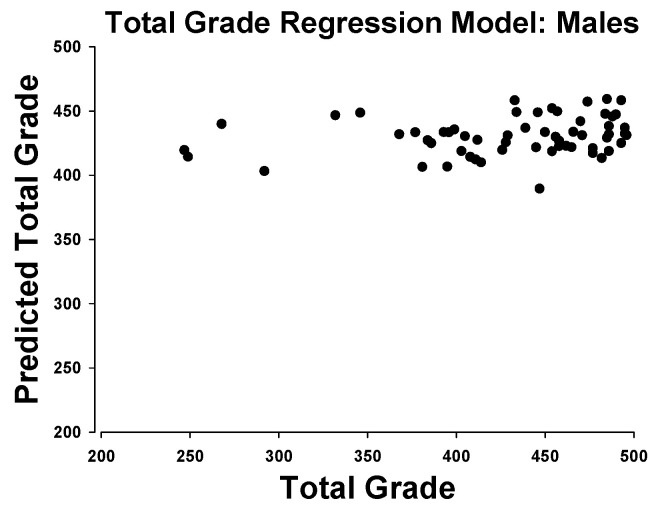
Actual total grade scores in males vs the predicted value from a regression model including only soda pop hand test times. No additional variables were able to improve the model in males.

**Table 1 ijerph-18-02293-t001:** Baseline anthropometrics and academic achievement data for the subjects.

		**Height (cm)**	**Weight (kg)**	**Age (years)**	**Body Fat (%)**	**BMI (kg/m^2^)**	**WHR (%)**
**Gender**	Male	140.5 ± 6.3	41 ± 10.7	8.9 ± 0.31	22.5 ± 11.1	20.6 ± 4.3	0.878 ± 0.08
	Female	141.9 ± 6.8	40.3 ± 10.6	9.0 ± 0.26	21.1 ± 6.1	19.8 ± 4.2	0.851 ± 0.064
		**Korean Lit. ***	**Math**	**Social Study**	**Science**	**English**	**Total Grade**
**Gender**	Male	89.3 ± 11.5	85.9 ± 15.1	87.7 ± 12.1	82.9 ± 14.6	84.1 ± 19.1	429.9 ± 60.6
	Female	94.4 ± 5.0	88.2 ± 10.1	89.9 ± 8.9	86.5 ± 10.6	89.3 ± 13.9	448.4 ± 36.7

Values are mean ± SD. BMI, body mass index; WHR, wait hip ratio. * denotes a significant difference between males and females at *p* < 0.05.

**Table 2 ijerph-18-02293-t002:** Physical activity and physical fitness test measures. Physical activity expressed as the daily average of a 7-day collection period.

		**PA (Steps) ***	**VO_2_max** **(mL/kg/min)**	**Sit Up * (Repetitions)**	**Sit and Reach * (cm)**	**Long Jump * (cm)**
**Gender**	Male	11,015 ± 2093	39.0 ± 9.8	17.2 ± 10.4	3.2 ± 17.3	123.9 ± 17.5
	Female	9745 ± 1503	39.7 ± 6.6	10.7 ± 7.1	8.5 ± 5.9	113.6 ± 13.3
		**50m Run (s)**	**Hand Grip (kg)**	**Soda Pop, Hand (s)**	**Soda Pop, Foot (s)**	**Illinois Agility * (s)**
**Gender**	Male	10.25 ± 1.41	19.2 ± 4.1	5.13 ± 0.59	14.00 ± 2.54	22.40 ± 1.88
	Female	10.20 ± 0.80	19.4 ± 4.2	5.33 ± 0.57	15.54 ± 2.55	23.97 ± 1.81

Values are mean ± SD. PA, physical activity; VO_2_max, maximal oxygen uptake. * denotes significant difference between males and females at *p* < 0.05.

## Data Availability

The data presented in this study are available on request from the corresponding author. The data are not publicly available due to the protection of personal information regarding the student’s grades.

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
