# Peer review of "The Associations between Physical Fitness, Complex vs Simple Movement, and Academic Achievement in a Cohort of Fourth Graders"

_ijerph, 2021, doi:10.3390/ijerph18052293_

Round 1
Reviewer 1 Report
The topic of The Associations Between Physical Fitness, Complex vs Simple Movement, and Academic Achievement in Elementary School Students is topical, as it concerns the problem of the occurrence of civilization diseases. This is due to from less and less willingly undertaken by children and adolescents physical activity, which translates into other areas of life, including science.
The subject of the study would fall under the section of the International Journal of Environmental Research and Public Health, if it largely related to the issues contained therein. However, the literature presented in the introduction does not deal with many health problems related to the lack of physical activity (including the current problems related to the pandemic - mental and social), what is more, pointing to many relationships between PA and e.g. cognitive functions was based on one item (line 45). There should also be an overview of the available and recommended physical fitness tests. The literature review should definitely be extended.
The research methodology is questionable. It is not known how the study sample was selected and how exactly the studies lasting the week were conducted. Why such fitness tests were chosen, and any factors that could interfere with the performance of the study, were not indicated.
The presented research results do not entitle to the presented general conclusions, for children from grades 4 of primary school - as the title suggests. In a way, this was indicated as a recommendation for future research. It is therefore worth mentioning in the title that it applies to students of a selected primary school.
In the discussion, the authors comparing their own results with other authors refer to studies conducted among preschool children (line 192, 193), while suggesting that motor coordination and the ability to maintain attention may also contribute to school success. This is too much of a conclusion (line 189), even if the requirement for further research is further emphasized (line 190).
Recommendation. Acceptance for publication after significant additions: clarification of the title, supplementation in the introduction with the literature on the subject, a more detailed description of the research methodology with argumentation for the selection of individual physical fitness tests. Limiting conclusions and translating research results into achieving school success, to a specific research group and specific correlations, and not to the entire population of grade 4 primary school students.
Author Response
We want to thank the reviewers for their time and commitment to review our manuscript. We appreciate the constructive criticism that was offered. We have addressed the raised concerns, and we believe that our manuscript has markedly improved as a result. We list each reviewer’s comments below in bold, followed by our reply.
Reviewer #1
- The topic of The Associations Between Physical Fitness, Complex vs Simple Movement, and Academic Achievement in Elementary School Students is topical, as it concerns the problem of the occurrence of civilization diseases. This is due to from less and less willingly undertaken by children and adolescents physical activity, which translates into other areas of life, including science.
The subject of the study would fall under the section of the International Journal of Environmental Research and Public Health, if it largely related to the issues contained therein. However, the literature presented in the introduction does not deal with many health problems related to the lack of physical activity (including the current problems related to the pandemic - mental and social), what is more, pointing to many relationships between PA and e.g. cognitive functions was based on one item (line 45). There should also be an overview of the available and recommended physical fitness tests. The literature review should definitely be extended.
We have brought additional information into the discussion to address this comment. Information was added to the intro (lines 43-49) highlighting the increased importance of these questions during the COVID pandemic, as well as additional literature regarding the links between PA and academics/cognition (57-61, 80-86).
In regard to the request for an overview of the available and recommended physical fitness tests, we do not feel that addressing this question can fit within the scope of the current work. However, the vast and disjointed possibilities when it comes to these assessments must be recognized. We have added in the introduction note of the fact that methods are not standardized (67-71), and in respect to this and subsequent comments, significant additions have been made to the methods to clarify test choices (113-135).
- The research methodology is questionable. It is not known how the study sample was selected and how exactly the studies lasting the week were conducted. Why such fitness tests were chosen, and any factors that could interfere with the performance of the study, were not indicated.
This point is well taken, and significant additions have been made throughout the methods section to clarify these questions, primarily lines 100-135.
- The presented research results do not entitle to the presented general conclusions, for children from grades 4 of primary school - as the title suggests. In a way, this was indicated as a recommendation for future research. It is therefore worth mentioning in the title that it applies to students of a selected primary school.
Title ending has been rephrased from “in Elementary School Students” to “in a Cohort of Fourth Graders” to reflect this limitation.
- In the discussion, the authors comparing their own results with other authors refer to studies conducted among preschool children (line 192, 193), while suggesting that motor coordination and the ability to maintain attention may also contribute to school success. This is too much of a conclusion (line 189), even if the requirement for further research is further emphasized (line 190).
We agree that this was too much of a reach. This section of the discussion (lines 253-269) has been rewritten to clearly indicate that while some previous correlations exist, the current dataset does not have the measures to make inferences in this regard. This comment has also led us to more clearly state these limitations in regards to the agility data as well (264-268).
- Acceptance for publication after significant additions: clarification of the title, supplementation in the introduction with the literature on the subject, a more detailed description of the research methodology with argumentation for the selection of individual physical fitness tests. Limiting conclusions and translating research results into achieving school success, to a specific research group and specific correlations, and not to the entire population of grade 4 primary school students.
We hope that we have sufficiently addressed these concerns through the changes noted above, as well as additional changes at 269-280 regarding the translation to achieving school success (for which, unfortunately, it is not yet possible to make mechanistic links) and at 338-340 clarifying the limitations inherent to extrapolating out from the current cohort.
We want to thank the reviewers for their time and commitment to review our manuscript. We appreciate the constructive criticism that was offered. We have addressed the raised concerns, and we believe that our manuscript has markedly improved as a result. We list each reviewer’s comments below in bold, followed by our reply.
Reviewer #1
- The topic of The Associations Between Physical Fitness, Complex vs Simple Movement, and Academic Achievement in Elementary School Students is topical, as it concerns the problem of the occurrence of civilization diseases. This is due to from less and less willingly undertaken by children and adolescents physical activity, which translates into other areas of life, including science.
The subject of the study would fall under the section of the International Journal of Environmental Research and Public Health, if it largely related to the issues contained therein. However, the literature presented in the introduction does not deal with many health problems related to the lack of physical activity (including the current problems related to the pandemic - mental and social), what is more, pointing to many relationships between PA and e.g. cognitive functions was based on one item (line 45). There should also be an overview of the available and recommended physical fitness tests. The literature review should definitely be extended.
We have brought additional information into the discussion to address this comment. Information was added to the intro (lines 43-49) highlighting the increased importance of these questions during the COVID pandemic, as well as additional literature regarding the links between PA and academics/cognition (57-61, 80-86).
In regard to the request for an overview of the available and recommended physical fitness tests, we do not feel that addressing this question can fit within the scope of the current work. However, the vast and disjointed possibilities when it comes to these assessments must be recognized. We have added in the introduction note of the fact that methods are not standardized (67-71), and in respect to this and subsequent comments, significant additions have been made to the methods to clarify test choices (113-135).
- The research methodology is questionable. It is not known how the study sample was selected and how exactly the studies lasting the week were conducted. Why such fitness tests were chosen, and any factors that could interfere with the performance of the study, were not indicated.
This point is well taken, and significant additions have been made throughout the methods section to clarify these questions, primarily lines 100-135.
- The presented research results do not entitle to the presented general conclusions, for children from grades 4 of primary school - as the title suggests. In a way, this was indicated as a recommendation for future research. It is therefore worth mentioning in the title that it applies to students of a selected primary school.
Title ending has been rephrased from “in Elementary School Students” to “in a Cohort of Fourth Graders” to reflect this limitation.
- In the discussion, the authors comparing their own results with other authors refer to studies conducted among preschool children (line 192, 193), while suggesting that motor coordination and the ability to maintain attention may also contribute to school success. This is too much of a conclusion (line 189), even if the requirement for further research is further emphasized (line 190).
We agree that this was too much of a reach. This section of the discussion (lines 253-269) has been rewritten to clearly indicate that while some previous correlations exist, the current dataset does not have the measures to make inferences in this regard. This comment has also led us to more clearly state these limitations in regards to the agility data as well (264-268).
- Acceptance for publication after significant additions: clarification of the title, supplementation in the introduction with the literature on the subject, a more detailed description of the research methodology with argumentation for the selection of individual physical fitness tests. Limiting conclusions and translating research results into achieving school success, to a specific research group and specific correlations, and not to the entire population of grade 4 primary school students.
We hope that we have sufficiently addressed these concerns through the changes noted above, as well as additional changes at 269-280 regarding the translation to achieving school success (for which, unfortunately, it is not yet possible to make mechanistic links) and at 338-340 clarifying the limitations inherent to extrapolating out from the current cohort.

Reviewer 2 Report
Overall reading this well-written paper was a pleasure. The question that remains with me is why does the regression explain 30.5% of the variance among females and 4.3% among males. This is a huge difference and reasons to explain this difference are not explained. The authors must make an effort at explaining why there is such large difference between the sexes. It makes me think that some extraneous variable or variables are not being accounted for. More explanation is needed. Also more detail is needed for the sample selection and whether parents provided infored consent. You report how many students participated, but not how many students were asked to participate. Also, what standard was used to declare data to be an outlier?
Author Response
We want to thank the reviewers for their time and commitment to review our manuscript. We appreciate the constructive criticism that was offered. We have addressed the raised concerns, and we believe that our manuscript has markedly improved as a result. We list each reviewer’s comments below in bold, followed by our reply.
Reviewer #2
- Overall reading this well-written paper was a pleasure. The question that remains with me is why does the regression explain 30.5% of the variance among females and 4.3% among males. This is a huge difference and reasons to explain this difference are not explained. The authors must make an effort at explaining why there is such large difference between the sexes. It makes me think that some extraneous variable or variables are not being accounted for. More explanation is needed. Also more detail is needed for the sample selection and whether parents provided infored consent. You report how many students participated, but not how many students were asked to participate. Also, what standard was used to declare data to be an outlier?
This is a very fair point—seeking explanation of one of the main findings. We have added significant new discussion on this question, lines 281-337. While it was not possible to come to definitive conclusions from the current data, we have done our best to address some possible reasons for this difference between genders.
Significantly more detail has also been added to the methods section (100-110, primarily) that will hopefully satisfactorily address the shortcoming in detail there. In regards to the outlier question, we must apologize that the statistical methods section somehow was chopped from the original submission, and we did not catch this. This section has been reinserted (151-159), and addresses this and other statistical questions.

Reviewer 3 Report
Ms. Ref. No.: ijerph-1107067 Title: The Associations Between Physical Fitness, Complex vs Simple Movement, and Academic Achievement in Elementary School Students The article method is poor and compromises the evaluation of the study, as well as the study design. Introduction The introduction is well written. The authors stated that “Although many studies suggest that PA is related to academic achievement, they are limited to aerobic exercise, amount of physical activity, and physical fitness and body composition”. The justification could be better explored by the authors, bringing more data about previous studies. In the hypothesis, the authors should point what the direction of the correlation among academic achievement (Korean literacy, mathematics, social study, science, and English) and physical activity. Materials and Methods Please include the research design in this section. A general improvement is needed in this section. More details (e.g. Physical activity was tracked by 1 week of 80 pedometer activity). The statistical analysis was not included in this section. Which tests were used? Considering the correlations and the Regression Modeling Discussion The discussion is superficial in an attempt to explain the relationship between the variables.Author Response
We want to thank the reviewers for their time and commitment to review our manuscript. We appreciate the constructive criticism that was offered. We have addressed the raised concerns, and we believe that our manuscript has markedly improved as a result. We list each reviewer’s comments below in bold, followed by our reply.
Reviewer #3
- Ref. No.: ijerph-1107067 Title: The Associations Between Physical Fitness, Complex vs Simple Movement, and Academic Achievement in Elementary School Students The article method is poor and compromises the evaluation of the study, as well as the study design. Introduction The introduction is well written. The authors stated that “Although many studies suggest that PA is related to academic achievement, they are limited to aerobic exercise, amount of physical activity, and physical fitness and body composition”. The justification could be better explored by the authors, bringing more data about previous studies. In the hypothesis, the authors should point what the direction of the correlation among academic achievement (Korean literacy, mathematics, social study, science, and English) and physical activity. Materials and Methods Please include the research design in this section. A general improvement is needed in this section. More details (e.g. Physical activity was tracked by 1 week of 80 pedometer activity). The statistical analysis was not included in this section. Which tests were used? Considering the correlations and the Regression Modeling Discussion The discussion is superficial in an attempt to explain the relationship between the variables.
Additional information (lines 57-61, 80-86) has been brought into the introduction to better explore the existing knowledge surrounding these relationships.
The directionality of the hypotheses has been clarified, lines 93-95. This is a good catch of an ambiguous setup.
Significant detail has been added to the methods section to clarify overall design, pedometer setup, statistical analysis, and more. On the statistical methods section especially we must apologize that this was somehow chopped from the original submission, and we did not catch this. In any case, we have clarified this and more in the methods, primarily lines 100-135 and 151-159.
While the relationships between the variables are not well understood, we have worked to flesh out these relationships with what is known in the discussion, attempting to better explain what inferences might be made as to the variable relationships (primarily 253-259, 264-280, 299-337).

Round 2
Reviewer 1 Report
Thank you for corrections, they improved the quality of the presentation.
Reviewer 3 Report
Significant changes were made. Accepted.